# Automated Identification and Localization of Rail Internal Defects Based on Object Detection Networks

**Sicheng Wang** [1,2,3] , **Bin Yan** [1,3] , **Xinyue Xu** [1,2,3], **Weidong Wang** [1,2,3], **Jun Peng** [1,2,3], **Yuzhe Zhang** [1,2,3], **Xiao Wei** [1,2,3] **and Wenbo Hu** [1,4,5,*]

1    School of Civil Engineering, Central South University, Changsha 410075, China;
     caesar.wang@csu.edu.cn (S.W.); binyan@csu.edu.cn (B.Y.); csuwwd@csu.edu.cn (W.W.);
     civilpengjun@csu.edu.cn (J.P.); 224811177@csu.edu.cn (Y.Z.); 204801066@csu.edu.cn (X.W.)
2    Key Laboratory of Heavy Haul Railway Engineering Structure, Ministry of Education,
     Changsha 410075, China
3    Research Center for Intelligent Monitoring of Rail Transit Infrastructure, Central South University,
     Changsha 410075, China
4    Department of Civil and Environmental Engineering, The Hong Kong Polytechnic University,
     Hong Kong 999077, China
5    National Rail Transit Electrification and Automation Engineering Technology Research Center (Hong Kong
     Branch), The Hong Kong Polytechnic University, Hong Kong 999077, China
*    Correspondence: wenbo.hu@polyu.edu.hk

**Abstract:** The timely identification of rail internal defects and the application of corresponding preventive measures would greatly reduce catastrophic failures, such as rail breakage. Ultrasonic rail defect detection is the current mainstream rail defect detection method thanks to its advantages of strong penetration, high accuracy, and ease to deploy. The 2D B-scan image output by ultrasonic detectors contains rich features of defects; however, rail engineers manually identify and localize the defect image, which can be time-consuming, and the image may be subject to missing identification or mistakes. This paper adopted state-of-the-art deep learning algorithms for novel B-scan images for the automatic identification and localization of rail internal tracks. First, through image pre-processing of classification, denoising, and augmentation, four categories of defect image datasets were established, namely crescent-shaped fatigue cracks, fishbolt hole cracks, rail web cracks, and rail base transverse cracks; then, four representatives of deep learning object detection networks, YOLOv8, YOLOv5, DETR, and Faster R-CNN, were trained with the defects dataset and further applied to the testing dataset for defect identification; finally, the performances of the three detection networks were compared and evaluated at the data level, the network structure level, and the interference adaptability level, respectively. The results show that the YOLOv8 network can effectively classify and localize four categories of internal rail defects in B-scan images with a 93.3% mean average precision at three images per second, and the detection time is 58.9%, 376.8%, and 123.2% faster than YOLO v5, DETR, and Faster R-CNN, respectively. The proposed approach could ensure the real-time, accurate, and efficient detection and analysis of internal defects to a rail.

**Keywords:** rail internal defect; ultrasonic detection; defect identification and localization; YOLOv8 network





## 1. Introduction

Rail tracks are one of the most critical infrastructure systems in the world. Repeated loads, the environment, and the quality of construction and material may introduce damage to rails, which could pose risks to their safe operation. One of the most concerning distresses is the internal defect of a rail (shown in Figure 1), which is challenging to be identified through visual inspection. Various methods for detecting these internal defects have been developed in the past, and significant progress has been made in defect identification algorithms and systems.

Inspection methods have developed from early traditional manual inspection to automatic inspection. At present, rail non-destructive automatic inspection systems, such as ultrasonic, magnetic flux leakage, eddy current, and non-destructive inspection using video camera, are widely used.

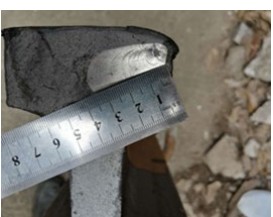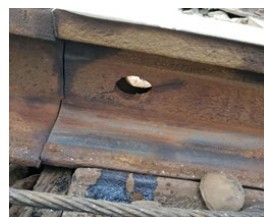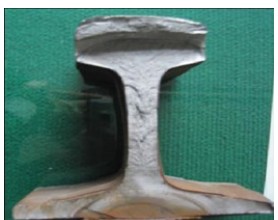

**Figure 1.** Examples of rail internal defects.

With the advancement of camera systems and the maturity of computer vision and image processing algorithms, the traditional manual inspection method, which has low-efficiency and is costly and subjective, has been gradually replaced. Automated image-collection vehicles with high-speed digital cameras, as the substitution, are installed on track checking trains to detect railway defects efficiently. Li et al. [1] proposed a real-time visual inspection system (VIS) to detect discrete rail surface defects, which can be detected in real time on a 216 km/h test train. Lee et al. [2] and Gong et al. [3] developed a system to acquire railway tunnel images using line scan cameras, which helped them realize the rapid detection of cracks in tunnel linings. Other studies [4–7], through extracting visual features based on the local spatial morphology of defects and through the use of texture analysis and intensity histogram analysis, detected rail surface defects automatically. However, the optical imagery caught by video camera may be prone to the illegibility of local information and a changing background due to shadow and light, and the images can only reflect the surface state of the rail, but not the internal state. As a result, other NDT (non-destructive testing) methods are also applied in this task, such as ultrasonic, eddy current, and magnetic flux leakage, and subsequently, intensive research has been directed to the field of defect assessment [8,9]. Sperry et al. [10] developed a non-destructive testing system for rail inspection; its inspection method involved magnetic induction, which could identify defects in the rail according to perturbations in the field. However, this technique is often considered cumbersome and sensitive, as electromagnetic noise and the presence of special parts might result in false detections. Eddy current testing [11] identifies defects using a magnetic field generated by eddy currents, and it has the common disadvantages of magnetic induction. The magnetic flux leakage (MFL) method is carried out through magnetic flux leaks from the rail wall in the location of a defect, which can be used for the evaluation of the rail's health condition. However, one shortcoming of MFL is the necessity of a rather intense magnetizing field. For the longitudinal magnetization of the rail, the magnetizing field is not virtually changed along the guide. Therefore, the method is typically used in car flaw detectors [12]. Compared with this, the ultrasonic method occupies a dominant position for rails in the field of rail inspection benefits from its excellent directivity, transmission, and reflection and refraction characteristics. Ultrasonic testing is reliable for detecting many deep surface-breaking cracks and internal defects in rails.

Based on ultrasonic testing, the existing methods are mainly used to detect rail defects through B-scan images acquired by an ultrasonic defect detector. The B-scan images could intuitively reflect the locations of the defects, which is beneficial for an operator to identify and locate them. Also, compared with optical imagery, ultrasonic B-scan images are not likely to be affected by light or shadows, and the resolution is fixed depending on the minimum scanning distance of the rail inspection vehicle, which avoids objects of the same category with different features due to different resolutions. Based on the characteristics of ultrasonic data, an SVM-based classification algorithm [13,14] was introduced to achieve the real-time detection and classification of rail defects. Li et al. [15] used an array probe to

send a linear equivalent modulation ultrasonic wave to a rail to detect internal defects, used the wavelet threshold to denoise the acquired signals, extracted features in the time domain and frequency domain, and used a support vector machine (SVM) to detect ultrasonic defects. Huang et al. [16] provided a BP neural network-based ultrasonic defect pattern classification technique to identify four common types of defects, and the effectiveness of the algorithm was verified with the example of bolt hole cracks. These machine vision-based ultrasonic processing methods saved labor and achieved good results. However, they relied on a large amount of prior knowledge and engineering experience to design the features learned. Due to the complexity and diversity of rail defects in shapes and orientations, and poor distinction between various defects, it is difficult to manually design accurate and robust feature descriptors for all rail defects, leading to stagnation in detection accuracy. Furthermore, the B-scan image is susceptible to clutter interference generated by the operating instrument, bringing greater impediments to the accuracy and efficiency of detection in machine learning.

In recent years, deep convolutional neural networks (CNNs) have been introduced to automatically extract features with good abstraction, accuracy, and robustness, and are widely used for image classification and recognition. Hu et al. [17] proposed an automatic classification method based on the ResNet-50 deep residual network for internal rail defects, which successfully classified four categories of defects in B-scan images. However, the category probability values as output were not intuitive. The method was unable to locate the defects from images and could not classify when multiple categories of defects appeared in one image. Benaissa et al. [18] successfully calculated the severity and crack depth of cracked beams by using vibration sensors based on the Teaching-Learning-Based Optimization Algorithm. Luo et al. [19] improved a new intelligent defect recognition system integrating deep learning and support vector machine, using a combination of deep separable convolution and selective search for target localization and support vector machine method for image classification, with the detection rate and accuracy of the ultrasonic B-scan image of defects higher than 95%. However, they used a combination of models that would increase the cost of (training and inference) time and cannot achieve end-to-end detection.

In pursuit of automating the localization and categorization of internal flaws in B-scan imagery, typically an object detection task, two primary framework types have emerged. The first is a dual-stage object detection approach, encompassing a region proposal network followed by a refined network. The alternative approach integrates object position regression with classification into a singular stage [20]. Among the two-stage methods, Faster R-CNN has demonstrated commendable accuracy, albeit with room for improvement in inference speed. Among them, the YOLO network is summarized as extremely fast, highly robust for near distance targets or small targets, and easy to be deployed on the mobile side [21]. Yuan et al. [22] introduced a YOLOv2-based recognition method combined with the Otsu algorithm, effectively identifying loose screws in the connection plates of low and medium speed maglev contact rails. Similarly, Wang et al. [20] conducted a comparative analysis of various one-stage deep learning methods for inspecting crucial railway track components such as rails, bolts, and clips, noting the YOLO models' superior speed at equivalent accuracy levels. Sikora P et al. [23] employed YOLO for the detection and classification of rail barriers at level crossings, including railway warnings and light signaling systems, achieving a mean average precision of 96.29%. Feng et al. [24] proposed a detection network incorporating a MobileNet backbone and novel detection layers, attaining high-accuracy detection and localization of rail surface defects. However, the application of object detection networks in internal defect detection of rails remains scarce, primarily focusing on optical images with limited usage in B-scan data processing [25–27]. The latest research [28] describes the application of the YOLO series in wound detection, which provides a good idea for our research.

In summary, the following problems remain unsolved on the internal defect detection of the rail. (1) For the existing NDT methods, the optical imagery taken by video cameras

cannot reflect the status of rail internal defects, and other NDT (such as Eddy Current and MFL) methods are sensitive and complex, which are unreliable in internal defect detection. (2) For the existing ultrasonic B-scan image processing algorithms, traditional machine vision-based algorithms (such as SVM) rely heavily on prior knowledge and engineering experience to design features, which is prone to accuracy stagnation; deep learning CNN-based methods, one is the classification algorithm cannot locate the defects, the second is the combinatorial algorithm would increase the inference time. (3) For object detection algorithm, most primarily used for processing optical images and very few was used for B-scan data. The latest research used it on B-scan images but has unclear classification criteria and no noise processing. To address the above issues, this paper based on YOLOv8, proposes to automatically classify and localize internal rail defects based on object detection models using B-scan images obtained by ultrasonic detector. The rest of this paper is organized as follows: Section 2 introduces the preparation and processing of datasets. Firstly, it comes up with a defect image classification principle according to the features of the B-scan image channel's color, morphology and relative location. Then, these images are pre-processed by denoising and data enhancement to establish an image dataset containing four categories of defects (head, bolt hole, web, base). Section 3 simply indicates the features of YOLO series, SSD and Faster R-CNN detectors, and then sets the YOLOv8 as an example to introduce the structural composition and improvement methods (small tricks) in the model. Section 4 transfer-learns four object detection networks and tests them on the test set, optimizes the parameter and validates generalization capability by adjusting hyperparameters and network details. Section 5 investigates the suitability of the YOLOv8 method in the test set, and further compares it with Faster R-CNN, DETR (Detection Transformer), and YOLOv5 in terms of the mean average precision (mAP), IOU, the *F*1 score, and inference time and discusses the results in different conditions, followed by the concluding remarks in Section 6.

## 2. Data Preparation

The preparation and processing of the B-scan image dataset consists of three steps. Firstly, the data is acquired by the GCT-8C rail defect detector and further comes up with a defect image classification (labeling) principle according to the features of the B-scan image channel's color, morphology and relative location. Then, these images are pre-processed to establish an image dataset containing four categories of defects (head, bolt hole, web, base).

### 2.1. Data Collection

The internal defects dataset in this research comes from the inspection records of railway lines in Nanchang, China. Figure 2 demonstrates how the rail defect detector was used to collect images of internal defects and the correspondence between B-scan images and actual defects.

For data processing, one of the most important conditions is acquisition devices and their parameters that determine the resulting quality. GCT-8C rail defect detector, a kind of small hand-pushed digital rail ultrasonic defect detector that has a total of nine detection channels: one 0° channel, two 37° channels, and six 70° channels ensuring sufficient view angle, used for detection. Each of the channels corresponds to a specific color (customized) and detection area, forming an image with complex colors but a regular layout. The main technical parameters of the instrument are shown in Table 1, and the setting of color and detection area of different channels are shown in Table 2. These channels run with a rate of 60 Hs per second along rail and display in 2 forms: ultrasonic A-scan image and B-scan image. In them, ultrasonic A-scan image is mainly used for on-site equipment status adjustment and monitoring due to the large data amount. In contrast, the ultrasonic B-scan image, which has a small amount of data, stores and further analyzes railway conditions. Furthermore, B-scan images could intuitively reflect the location of the defects, which is beneficial for the operator to identify and locate them. Also, compared with traditional images, ultrasonic B-scan images are not likely to be affected by light or shadows, and the

resolution is fixed, depending on the minimum scanning distance of the rail inspection vehicle, which avoids objects of the same category with different features due to different resolutions. In this study, the collected continuous long B-scan images were cut and extracted to establish a dataset containing 200 samples, and each sample has a pixel size of 1325 pixels × 346 pixels.

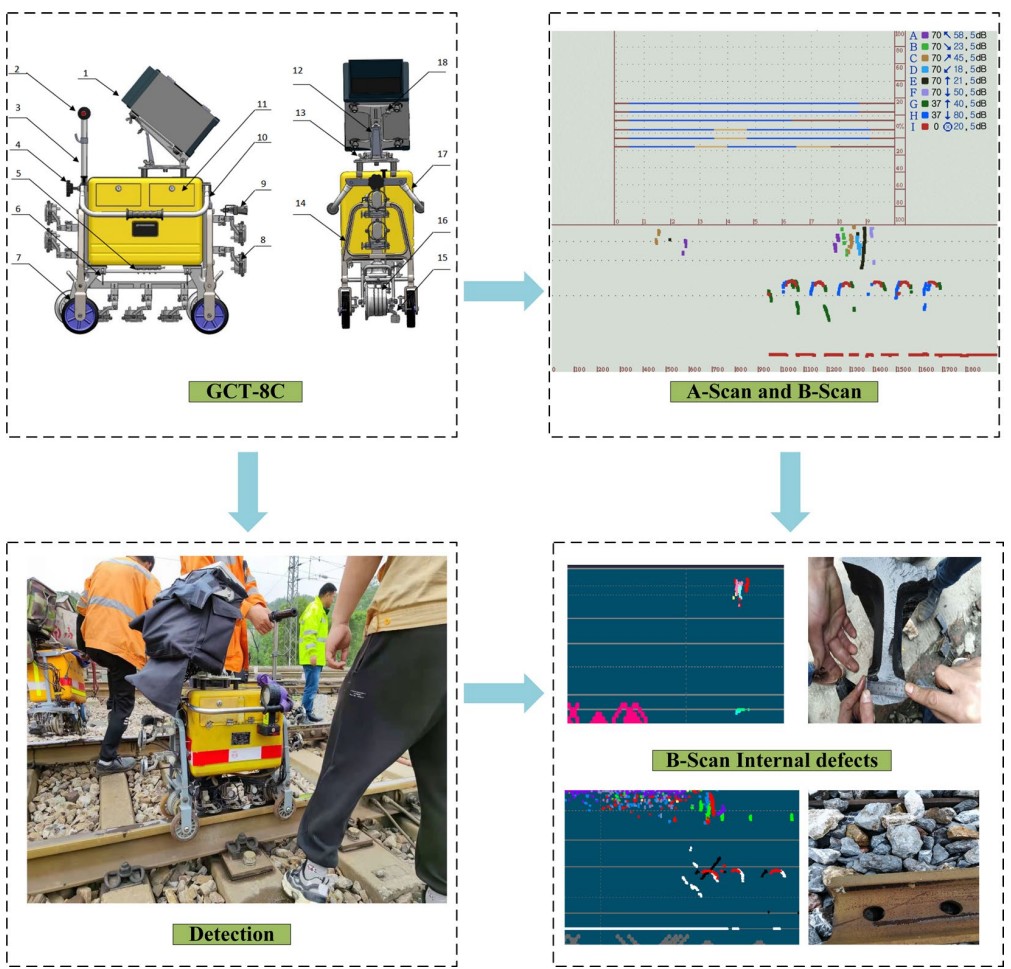

**Figure 2.** The structure of GCT-8C rail defect detector.

**Table 1.** Main technical parameters.

| Classification | Parameter |
| --- | --- |
| Channel Number | 9 |
| Swipe speed/(Hz·s$^{-1}$) | 60 |
| Probe operating frequency/Hz | 400 |
| Repetition detection frequency/Hz | 400 (per channel) |
| Detection range/mm | 0~200 ± 4 (43 kg/m and 50 kg/m rail); 0~250 ± 5 (60 kg/m and 75 kg/m rail) |
| Horizontal linearity error | ≤2% |
| Vertical linearity error | ≤5% |

However, the established B-scan image dataset has some drawbacks: the data is extremely sparse, lacks features, and is always noisy. Due to the characteristics of B-scan data, the corresponding image processing methods are also specific, and to address these drawbacks, pre-processing methods are introduced in the next steps.

**Table 2.** The properties of different channels.

| Probe Angel | Channel Name | Channel Color (Customized) | Detection Area |
|---|---|---|---|
| 70° | ABCDEF | dark pink, light green, brown, cyan, black, light pink | head and fillet; |
| 37° | GH | green, blue | web and base; |
| 0° | I | red | web and its projection area |

*2.2. Internal Defects Classification from Collected B-Scan Images*

The ultrasonic B-scan image is a kind of continuous 2D image data that can display the cross-sectional image of the detected object. In B-scan images, the horizontal coordinates represent the probe movement distance, and the vertical coordinates represent the acoustic wave propagation time or distance. This intuitively reflects the detailed distribution in the probe detection range and the depth of the detected object from the detection surface.

Based on the ultrasonic special transmission, reflection, and refraction characteristics of ultrasonic, when encountering a discontinuous medium, the wave will reflect and represent different colors and morphological characteristics. These discontinuous structures that can reflect the ultrasonic wave and can be received by the echo sensor are called ultrasonic reflector. The ultrasonic reflector includes both normal rail structures, such as screw holes, conductor holes, and rail joints, and rail defects, such as fishbolt hole cracks, crescent-shaped fatigue cracks, and rail base transverse cracks. According to the wave distribution regular, rail defects were first picked out from the ultrasonic reflectors. Then, the selected rail defects are further subdivided. The ultrasonic defects were manually labeled to fully learn these ultrasonic B-scan rail defect image features in the following training stage. Due to the lack of features in the B-scan images dataset and because the morphology of different categories of defects B-scan images are similar, this paper proposes a three-step classification method: the first step is a coarse classification method based on the colors and morphological characteristics of the channels, the second step is a fine classification method based on geometrics moments, and the third step is to frame in auxiliary line information during the labeling process to add location features. The specific classification method based on B-scan image features is divided into three parts, which are as follows:

2.2.1. Distinction of Defects Based on Colors and Morphological Characteristics

The three boundary lines in the B-scan images divide the rail cross-section into three parts: rail head, rail web, and rail base, as shown in Figure 3a. The corresponding area in the ultrasonic B-scan image is shown in Figure 3b. When the reflectors appear in different parts, the channel displays different colors and morphological characteristics, which represent the category and extent of the defects: When reflectors are located in the rail head or rail fillet areas, channels of reflectors are shown as monochrome or multi-color. In this area, the reflectors are likely to belong to two situations: a weld joint which is a normal structure, and the other is rail head nuclear injury, which is a defect. The key to distinguishing these two situations is the channel color of the straight 70° probe; when the color appears, the reflector is nuclear injury; when reflectors are located in the rail web areas, channels show a monochrome or combination of red, green and blue. In this area, the reflectors will likely be normal bolt holes or the cracks around the hole. If there are cracks, the channels of the upper cracks and lower cracks of the hole are regularly shown as green and blue, respectively, and the horizontal crack is shown as red. When reflectors in the rail base area, channels show a monochrome or combination of red, green, and blue. If there are rail base transverse cracks, the channels of the horizontal crack of the rail base are blue and green, and the longitudinal crack are red. Through this way, the rail internal defects have been coarsely classified. The classification of rail defects and the representatives of B-scan images of each category are shown in Table 3.

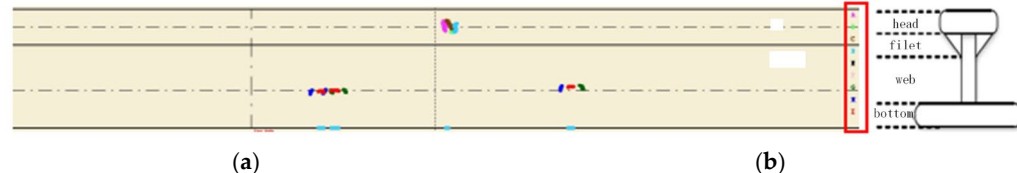

|     (**a**)     |     (**b**)     |

**Figure 3.** (**a**) Rail cross-section image. (**b**) Ultrasonic B-scan image in the corresponding area.

**Table 3.** The classification of rail defects and the representatives of B-scan images.

| Rail Defects | B-Scan Image |
| --- | --- |
| crescent-shaped fatigue cracks | |
| fishbolt hole cracks | |
| rail web cracks | |
| rail base transverse cracks | |

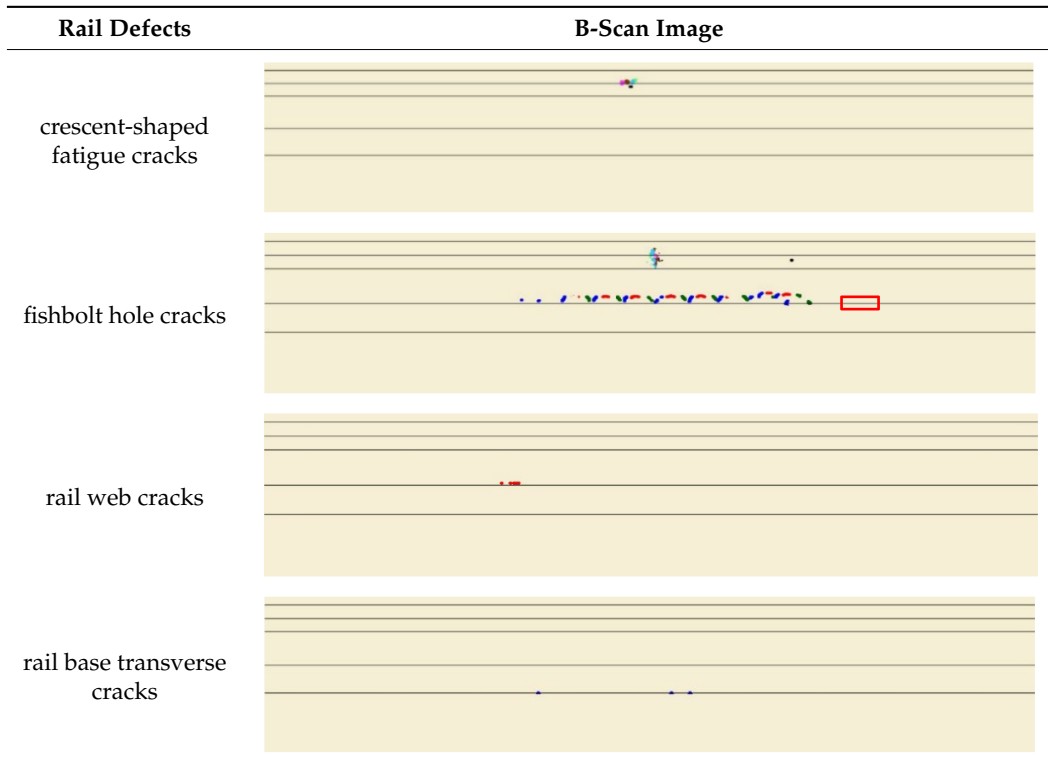

However, when reflectors are located in the rail web areas, the complex reflection might lead to a confusing distribution of channels, which cannot be easily distinguished by the above method.

### 2.2.2. Distinction of Defects Based on Geometric Moments

Holes including bolt holes, wire guide holes and structure holes, are normal ultrasonic reflectors. Its B-scan imaging is difficult to distinguish from that of holes surrounding cracks. It is shown in Figure 4. Wu et al. [29] analyzed and extracted features of every rail defect, including color features, distribution features, and contour features, and built the numerical constraint relationship of the different defect categories, which solved this problem. Based on color and distribution features, the B-scan image of a bolt hole is displayed by intersecting green-red-blue channels, and the channels of bolt hole cracks are located on both sides of that bolt hole. Generally, the channels represent the upper cracks of a bolt hole are typically positioned on either side of the bolt hole's channel, with the green channel commonly situated to the right and the blue channel to the left of the bolt hole channel. When cracks adjacent to the bolt hole are absent, the channels of identical hues associated with the bolt holes exhibit a pattern of equidistant arrangement along the horizontal axis. Conversely, the presence of cracks in the vicinity of the bolt hole disrupts this equidistant channel distribution. Concerning the lower cracks of the bolt hole, they are categorized into two distinct formations, separated lower screw cracks and bonded lower screw cracks, which need numerical constraints. The contour moments are calculated by

integrating over all pixel points on the contour boundary. For a $M \times N$ size of image, when its function is $f(i, j)$, the calculation of $p + q$ order geometric moment $m_{pq}$ is as follows:

$$m_{pq} = \sum_{i=1}^{M} \sum_{j=1}^{N} i^p j^q f(i, j) \tag{1}$$

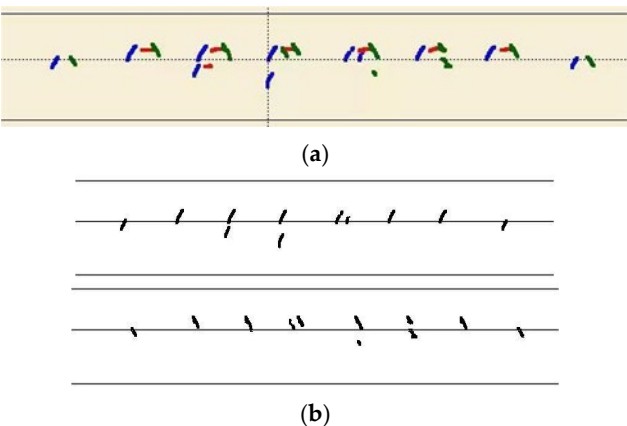

(a)

(b)

**Figure 4.** (**a**) The B-scan image containing two kinds of bolt hole cracks. (**b**) The binary images of blue image channel and green image channel.

Then, calculate the 0th-order moment and 1st-order moment of $m_{pq}$, respectively. When the image is a binary image, the 0th-order moment gains the area of the image contour and the 1st-order moment gains the coordinates of the center of gravity of the image $(\bar{i}, \bar{j})$. The specific calculation is as follows:

$$m_{00} = \sum_{i=1}^{M} \sum_{j=1}^{N} f(i, j) \tag{2}$$

$$m_{10} = \sum_{i=1}^{M} \sum_{j=1}^{N} i \cdot f(i, j) \tag{3}$$

$$m_{01} = \sum_{i=1}^{M} \sum_{j=1}^{N} j \cdot f(i, j) \tag{4}$$

$$\bar{i} = \frac{m_{10}}{m_{00}}, \ \bar{j} = \frac{m_{01}}{m_{00}} \tag{5}$$

Based on contour features, the horizontal distance between the center of gravity of the upper crack channel and the center of gravity of the bolt hole channel is greater than its vertical distance, that is $\Delta\bar{i} > \Delta\bar{j}$. And the rule for lower crack is the opposite, that is $\Delta\bar{i} < \Delta\bar{j}$.

### 2.2.3. Label Division

The categories of rail defects are different, and the hazards and maintenance methods required are also different. Normally, the defects originate from the manufacturing process, cyclical loading, impact from rolling stock, rail wear, and plastic flow [30]. Refer to the "Railway Defect Classification" (TB/T 1778-2010 [31]). In this paper, the dataset is labeled into four categories of rail defects: crescent-shaped fatigue cracks (head), fishbolt hole cracks (bolt hole), rail web cracks (web) and rail base transverse cracks (base), and their counterparts, normal rail structure. Each image with a pixel size of 1325 pixels × 346 pixels usually contains one or more labels. According to the 9:1 principle of training set data

and test set data, 20 images (containing 16 labels) are taken as the test set, and 180 images (containing 176 labels) are taken as the training set, as shown in Table 4.

**Table 4.** Labels and quantity of B-scan images dataset.

| Image Category | Label | Training Set | Test Set | Total |
|---|---|---|---|---|
| crescent-shaped fatigue cracks | head | 96 | 2 | 98 |
| fishbolt hole cracks | bolt hole | 22 | 5 | 27 |
| rail web cracks | web | 30 | 7 | 37 |
| rail base transverse cracks | base | 28 | 2 | 30 |
| total | | 176 | 16 | 192 |

*2.3. Data Pre-Processing*

The dataset in this research has the problems of the data being sparse and the image scale of the four labels being uneven. Most of the rail defects are concentrated in the rail head area, with 96 and 79 images of crescent-shaped fatigue cracks and scaling, respectively. This paper uses data enhancement to expand the image set of the web and base area. The category of target defect in B-scan images is related to the position of the whole image, which limits the enhancement way. So, data enhancement mainly uses random mirror transformation (only left and right) and random miscut transformation to change each image while randomly maintaining the same pixel size. In this way, the sizes of the original image set of fishbolt hole cracks, rail web cracks, and rail base transverse crack expand three times, reaching a balance of all defect categories. It is conducive for the network to learn image feature information fully.

The last problem of the dataset is always noisy. Unlike the Gaussian white noise in ordinary images, the noise in ultrasonic B-scan images is influenced by complicated factors during the ultrasonic wave propagation process, including the state of the rail surface, water coupling, electronic noise, and impacts of detection parameters. A B-scan image is a two-dimensional image created by superimposing the pixel values on the corresponding position of the three-primary-color two-dimensional matrix. The pixel (echo) points of the same reflector are closely connected. This paper adopts the pixel subtraction method and eight-directional point finding denoising method, that is, starting from one-pixel point, using it as the starting point to find consecutive non-background color pixel points from each of the eight directions and add up these pixel points, and if the number of consecutive pixel points is less than or equal to 3, it will be regarded as a spurious wave, that is noise. This paper eliminates noise by setting the threshold of the echo points number as 3 to build a cleaner and clearer dataset. Figure 5a shows the schematic diagram eight-point direction finding denoising method. Figure 5b shows the comparison of B-scan images before and after denoising.

Finally, 20 images (containing 16 labels) were taken as the test set and 282 images (containing 336 labels) as the training set. Table 5 shows the labels and quantity of the pre-processed B-scan image dataset.

**Table 5.** Labels and quantity of pre-processed B-scan images dataset.

| Image Category | Label | Training Set | Test Set | Total |
|---|---|---|---|---|
| crescent-shaped fatigue cracks | head | 96 | 2 | 98 |
| fishbolt hole cracks | web1 | 66 | 5 | 71 |
| rail web cracks | web2 | 90 | 7 | 97 |
| rail base transverse cracks | base | 84 | 2 | 86 |
| total | | 336 | 16 | 352 |

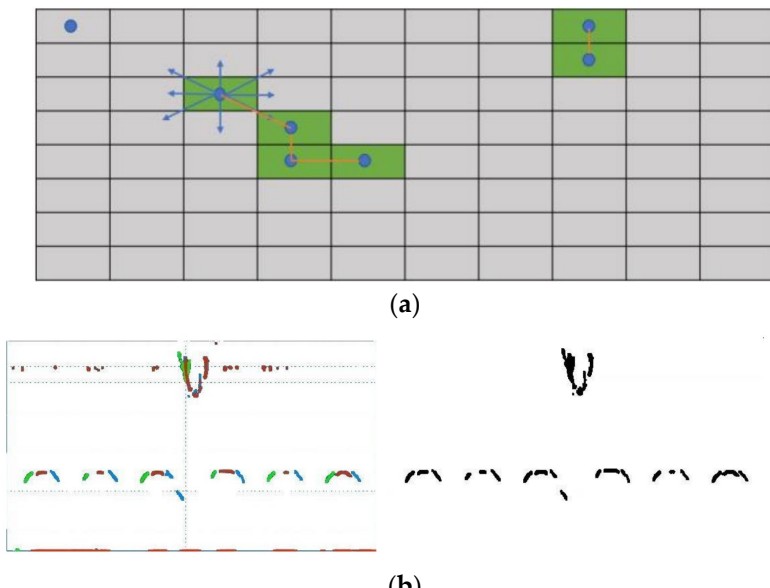

(a)

(b)

**Figure 5.** (**a**) Eight-point direction finding denoising method. (**b**) Denoising effect diagram.

## 3. Methodology

The objective of this study is to develop an automated methodology for the classification and precise localization of internal rail defects, leveraging advanced object detection algorithms applied to B-scan image datasets. The architecture of the object detection network is tripartite, encompassing the backbone, neck, and head components.

The backbone is a fundamental feature extractor, processing input images or videos to produce pertinent feature maps [32]. The backbone selection is pivotal, emphasizing optimizing a balance among accuracy, computational speed, and operational efficiency. Sophisticated, highly interconnected backbones such as ResNet and DenseNet offer enhanced accuracy but at the expense of increased processing time. Conversely, streamlined architectures like MobileNet and EfficientNet are preferred for their efficiency and swifter processing capabilities, providing a pragmatic trade-off between speed and performance accuracy [33]. The neck component of the network amalgamates diverse feature maps, with prominent examples including the Spatial Pyramid Pooling (SPP), Path Aggregation Network (PAN), and Feature Pyramid Network (FPN). The head component is responsible for the prediction phase, comprising two distinct types of detectors: the one-stage and the two-stage detectors. The two-stage detector incorporates a Regional Proposal Network (RPN) and a Region of Interest (RoI) pooling network. In this setup, the RPN layer forwards region proposals to a classifier and regressor for classification and bounding box regression. The one-stage detector, in contrast, eschews the region proposal phase, directly predicting bounding boxes from the input images and conflating location regression and classification into a singular, streamlined process. While two-stage networks such as Fast R-CNN and Faster R-CNN have achieved notable accuracy in localization and recognition, their application in real-time scenarios like pedestrian detection and video analysis is often limited by prolonged inference times [34]. To address these limitations, one-stage networks like the YOLO series and SSD have been developed, characterized by their high inference speeds and aimed at establishing efficacious real-time object detection systems [9].

### 3.1. YOLO Series Detectors

YOLO [35] (you only look once), a seminal one-stage object detection model, was conceptualized by Redmon and colleagues. It stands out for its capability to directly ascertain bounding boxes, associated confidence levels, and class probabilities of objects within input images. Notably, YOLO generates a considerably reduced count of bounding boxes per image compared to Faster R-CNN, facilitating an end-to-end real-time detection

process. An enhanced iteration, DETR [36], has emerged as a prominent model, acclaimed for its exemplary balance of processing speed and detection accuracy, securing its position as one of the most prevalently utilized deep learning frameworks globally. The DETR algorithm represents another significant innovation in object detection, employing the Transformer architecture as opposed to the reliance on convolutional neural networks seen in models like YOLOv8. By utilizing self-attention mechanisms, DETR can process images holistically, achieving commendable results. However, a notable drawback of DETR, as evidenced in our computations, is its slower operational speed compared to CNN-based approaches such as YOLOv8. This aspect underscores the trade-off between the comprehensive contextual analysis afforded by the Transformer model and the expedited processing characteristic of CNN-based detectors.

The latest YOLO version, v8, has shown significant improvements in the accuracy and speed of deep learning for object detection, particularly in terms of accuracy and speed in defect detection [37,38]. YOLOv8, while maintaining a comparable parameter count, achieves higher throughput, signifying a substantial progression in deep learning applied to object detection. This development has propelled YOLOv8 to the forefront of this research, surpassing previous versions of YOLO.

YOLOv8 inherits and enhances the core attributes of the YOLO series, including its capacity to directly predict bounding boxes, confidence levels, and class probabilities from input images. Compared to Faster R-CNN, YOLOv8 generates significantly fewer bounding boxes per image, facilitating an end-to-end real-time detection process. Moreover, YOLOv8 introduces innovations in structure and performance, employing a novel network architecture and advanced feature extraction methodologies. Notably, YOLOv8 adopts a new backbone network structure and an anchor-free detection head, leading to heightened detection precision, augmented multi-scale detection capabilities, and more efficient feature fusion. These improvements ensure the acquisition of rich semantic information while enhancing multi-scale detection ability and feature fusion effectiveness. YOLOv8 also incorporates new loss functions, including classification loss, bounding box regression loss, and IOU balance loss, optimizing model performance during training.

Recent advancements in YOLOv8 encompass training and network optimization techniques [39], such as data augmentation, learning rate adjustments, and selecting activation functions, further enhancing the model's performance across various scenarios. This is evident in the rail defect B-scan image inspection process depicted in Figures 6 and 7 and the comprehensive architecture of YOLOv8.

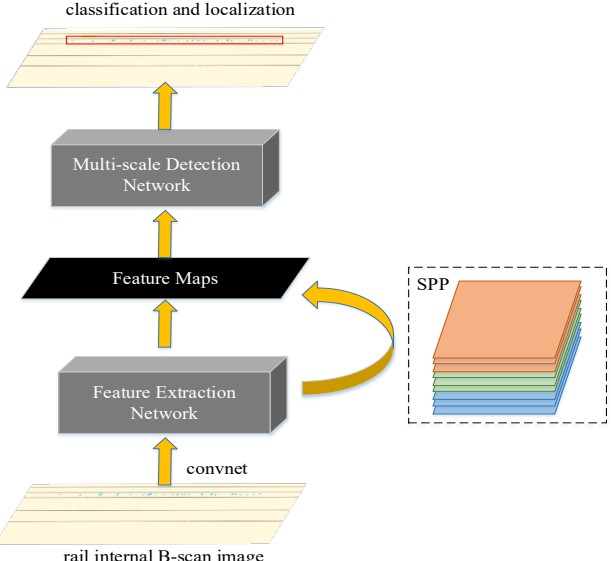

**Figure 6.** Overall framework for rail internal defect B-scan image detection.

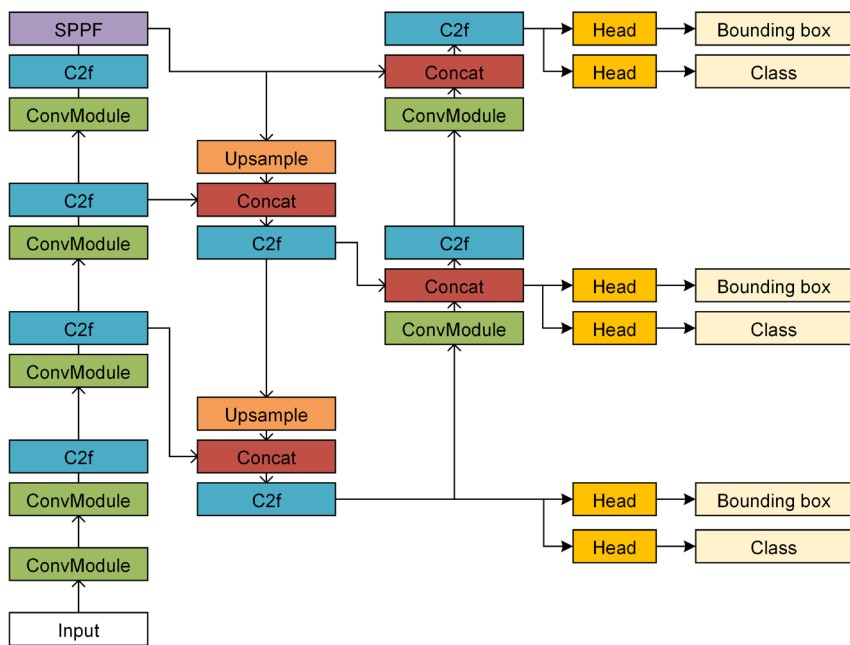

**Figure 7.** The architecture of YOLOv8 used in this study.

### 3.2. Other Object Detectors

#### 3.2.1. SSD

The Single Shot MultiBox Detector (SSD) [40], represents an innovative one-stage detection framework capable of simultaneously identifying multiple categories. It uniquely determines categorical scores and offsets for bounding boxes across various scales. Predominantly, SSD utilizes the VGG-16 architecture [41] as its backbone. Feature computation across individual image scales occurs independently within each feature map, from the highest to the lowest layers. During the training phase, anchor bounding boxes are aligned with corresponding ground truth boxes, classifying matches as positive instances and non-matches as negative. Empirical analyses demonstrate that SSD achieves robust performance metrics, particularly in the mean average precision (mAP) and processing speed.

#### 3.2.2. Faster R-CNN

Faster R-CNN [42] stands out as a particularly noteworthy model among the plethora of two-stage detectors. Typically employing ResNet50 [43] or VGG16 as its backbone, Faster R-CNN processes the extracted feature maps in two stages. The region proposal network (RPN) generates candidate boxes across three scales and three aspect ratios. These anchors are then classified as positive or negative using a Softmax function, and their positions are refined through bounding box regression to yield precise proposals. In the subsequent stage, the RoIPool (Region of Interest Pooling) operation extracts input feature maps and proposals from each candidate box, facilitating the tasks of classification and bounding-box regression. The model then calculates the proximity between the predicted box and the corresponding ground truth, optimizing the predicted box's location. Empirical results underscore that Faster R-CNN has significantly enhanced both accuracy and efficiency compared to its predecessors.

### 3.3. Object Detection Networks for Rail Defects Detection

Input B-scan images into the object detection network. In the training stage, the anchor bounding box (bbox) is used as the training sample, and find the anchor bbox, which has the largest IOU with each ground truth bbox, then use the label of the ground truth bbox as labeled of the anchor bbox, and then further calculates the offset of the anchor bbox relative to the ground truth bbox. The final purpose is to train the parameters of the anchor bbox to fit the ground truth bbox. Multiple anchor bboxes are first generated in the image in

the prediction stage. Then, the category and offset of these anchor bboxes are predicted according to the trained model parameters to obtain the prediction bbox. The model will output the probability value of each category, and the category with the highest probability value is used as the category of the prediction bbox. In this way, the classification and localization of rail defects from the B-scan image dataset.

## 4. Experiment

### 4.1. Experimental Environment and Strategies

Pytorch1.5.0, developed by Facebook, is used to construct the rail defects detection model. To accelerate the operation of the model with RTX 3090 GPU, CUDA 10.2 and Cudnn 7.6.5, which are developed by Nvidia, are also deployed in the computing environment. The object detection algorithms were written in Python 3.6.0.

Deep learning requires a large amount of high-quality labeled data, and pre-training with fine-tuning is now a very popular skill (trick) in deep learning, especially in the image inspection field as the representative. Pre-training is to use the pre-trained model as a feature extractor; in many cases, we choose pre-trained ImageNet, which includes 1.2 million images with 1000 labels, to initialize the model. Then, we change the fully connected layer to adapt specific classification tasks for output, or remove the last layer and replace it with a specific classifier, and retain the rest of the network structure as a feature extractor. Due to the small size of the rail defects B-scan dataset, this paper, through transfer learning [44], transfers the structure and weight parameters to rail defects classification, and the last classification layer of the network is modified to output the four category labels, which improves the training speed and generalization performance.

$B_p$ is the predicted bounding box, and $B_g$ is the ground truth bounding box.

Generally, an IOU threshold is predefined (the IOU threshold is traditionally set by 0.5). If the IOU value is greater than a predefined threshold, the predicted bounding box is determined to be a positive sample; otherwise, it is a negative sample. If the precondition that the classification result is correct or incorrect is also met, respectively, the above two situations are defined as true positive (TP) and false positive (FP) correspondingly. The other two metrics are false negative (FN) and true negative (TN). These four calculate the precision, recall, and *F*1 score values. The calculation method is shown in Equations (6)–(8).

$$Precision = \frac{TP}{TP + FP} \tag{6}$$

$$Recall = \frac{TP}{TP + FN} \tag{7}$$

$$F1 = \frac{2 \cdot Precision \cdot Recall}{Precision + Recall} \tag{8}$$

Precision represents the ratio of correctly identified positive instances to all positive predictions made by the model. Recall measures the proportion of true positives relative to the entire set of actual positive cases. The *F*1 score, a harmonic mean of precision and recall, is calculated at specific threshold levels [19]. The mean average precision (mAP) assesses the comprehensive performance of the model, taking into account varying Intersection over Unions (IOUs). This metric is derived from the average of the average precision (AP) for all detected object types, with AP being determined by the area under the precision–recall (PR) curve. This study establishes the IOU threshold at 0.5, with the corresponding PR curve depicted in Figure 8b. Additionally, inference time denotes the model's efficiency in processing images per second, implying that higher processing speeds signify superior algorithmic performance. The algorithm's complexity is quantified in terms of billion floating-point operations (BFLOPs) accumulated across multiple convolutional processes. On a specified GPU, inference time is inherently associated with BFLOPs [45].

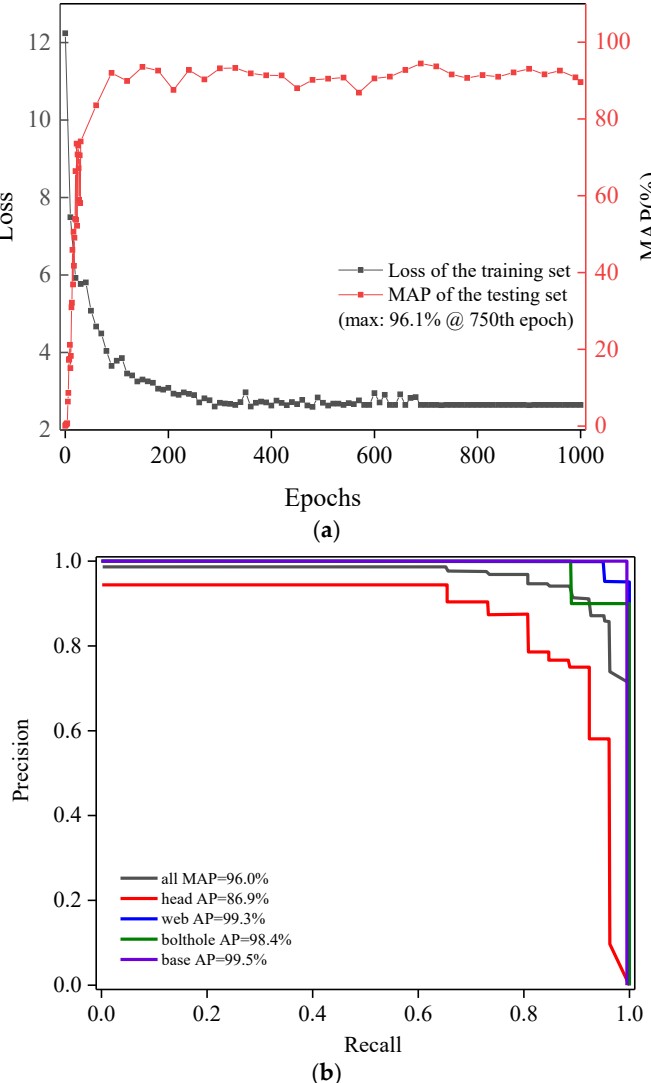

**Figure 8.** (**a**) mAP and loss–epochs curves of the YOLOv8 model. (**b**) YOLOv8 PR curve.

*4.2. Hyperparameters*

A comprehensive series of parameter optimization experiments were conducted on a set of 340 training images and 20 test images to ascertain the most effective hyperparameters, as delineated in Table 6. Given the model's convergence with the original dataset via transfer learning, a reduced learning rate of 0.0001 was set to fine-tune the rail defect B-scan image data. Employing a cosine annealing approach, the learning rate was reduced from the 8000th to the 9000th epoch, with the total epochs set at 10,000. Batch size, a critical parameter correlating with training duration and GPU capacity, was configured at 16 to maintain an equilibrium between these factors. Models including YOLOv8, DETR, YOLOv5, and Faster R-CNN were employed for the training process. To facilitate a fair comparison, the image input size for all models was standardized at 608 × 608 pixels. The primary objective of the training was to optimize model performance by minimizing the loss function.

**Table 6.** Training hyperparameters of the model.

| Hyperparameters | Values |
| --- | --- |
| Input size | $608 \times 608$ |
| Initial learning rate | 0.0001 |
| Learning rate decay | 0.8–0.9 |
| Momentum factor | 0.9 |
| Epochs | 10,000 |
| Batch size | 16 |

The bounding box regression loss typically employs mean squared error (MSE) for direct regression on the bounding's box coordinates, height, and width [46], with binary cross-entropy loss applied to analyze confidence and classification losses. Compared to early YOLO, YOLOV5 innovates in bounding box regression by replacing MSE with CIOU [47]. Further advancements in YOLOv8 include the substitution of CIOU with GIOU [48], which optimizes boundary overlap, non-overlap issues, center distance, and aspect ratio scaling, as detailed in Equations (9) and (10).

$$
\begin{aligned}
L_{CIOU} &= 1 - IOU(A, B) + \frac{\rho^2(A_{ctr}, B_{ctr})}{c^2} + \alpha \cdot v \\
v &= \frac{4}{\pi^2} \left( \arctan\frac{\omega^{gt}}{h^{gt}} - \arctan\frac{\omega}{h} \right)^2 \\
\alpha &= \frac{v}{(1 - IOU) + v'}
\end{aligned}
\tag{9}
$$

Within this framework, $\omega^{gt}$ and $h^{gt}$ symbolize the width and height of the ground truth bounding box, respectively. Conversely, $\omega$ and $h$ correspond to the width and height of the predicted bbox. Additionally, $\alpha$ and $v$ are penalty parameters related to the aspect ratio, where $\alpha$ is a positive scalar and v quantifies the aspect ratio consistency.

$$
GIOU = IOU - \frac{Area\,C - Area\,A \cup B}{Area\,A \cup B}
\tag{10}
$$

In this context, $Area\,A \cup B$ refers to the combined area encompassed by A and B. The Intersection over Union ($IOU$) metric remains at 0 in scenarios where boxes A and B do not overlap. Here, C signifies the smallest rectangle that completely encases both boxes A and B, with $Area\,C$ indicating the total area covered by rectangle C.

*4.3. Training Results*

The dataset is segregated into a training set and a test set. The former is utilized for the calibration of the model's weight parameters, while the latter assesses the model's efficacy.

Figure 8a illustrates the mAP and loss trajectory of the YOLOv8 model, configured with an input resolution of $608 \times 608$ pixels. Initially, the loss value is relatively elevated due to a higher learning rate. As the number of training epochs progresses, there is a gradual decrement in the loss value, culminating in convergence and stabilization at a near-constant value post 3000 epochs. Regarding model fitting, the apex of accuracy on the training set was observed at the 6000th epoch, achieving a proficiency level of 88%.

*4.4. Test Results and Evaluation Metrics*

After each epoch is trained, validation is carried out on the test set. The loss of the test set is approximately equal to that of the training set in the end, so there is no need to deal with overfitting. The rail defects detection model achieved outstanding performance in the test set, the results of four different categories defects (detect by YOLOv8, YOLOv5, DETR and Faster R-CNN). To assess the test performance visually, evaluation of the detection efficacy is conducted using three widely recognized metrics: the mean average precision (mAP), the *F*1 score, and the duration of inference. A critical aspect of mAP, the Intersection

over Union (IOU), quantifies the extent of overlap between the predicted outcomes and the actual ground truth bounding boxes, formulated as follows:

$$IOU = \frac{Area\left(B_p \cap B_g\right)}{Area\left(B_p \cup B_g\right)} \tag{11}$$

## 5. Results and Discuss

### 5.1. Model Comparison

The representatives of one-stage approaches, the DETR model, the YOLOv5 model, and the YOLOv8 model, and the representative of two-stage approaches, Faster R-CNN model, are trained with hyperparameters set. When the IOU threshold is set at 0.5 and the confident threshold is set at 0.5, the PR curve of different categories of rail defects based on these four models is shown in Figure 9. The *AP*, the *F1* score, and inference time results of each category for all test samples are listed in Table 7.

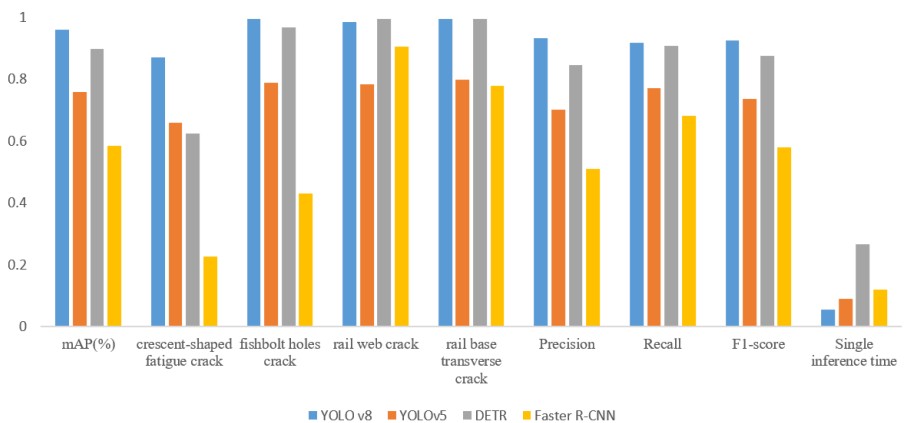

**Figure 9.** Performance comparison of different models.

**Table 7.** Performances on testing dataset algorithms.

| Model | mAP (%) | AP (%) | | | | Threshold = 0.5 | | | |
| --- | --- | --- | --- | --- | --- | --- | --- | --- | --- |
| | | Crescent-Shaped Fatigue Cracks | Fishbolt Hole Cracks | Rail Web Cracks | Rail Base Transverse Cracks | Precision | Recall | F1 Score | Single Inference Time (s) |
| YOLO v8 | 0.96 | 0.869 | 0.993 | 0.984 | 0.995 | 0.933 | 0.916 | 0.924 | 0.056 |
| YOLOv5 | 0.757 | 0.659 | 0.789 | 0.783 | 0.799 | 0.702 | 0.771 | 0.735 | 0.089 |
| DETR | 0.896 | 0.625 | 0.968 | 0.995 | 0.995 | 0.845 | 0.908 | 0.875 | 0.267 |
| Faster R-CNN | 0.585 | 0.227 | 0.429 | 0.905 | 0.778 | 0.510 | 0.680 | 0.580 | 0.125 |

YOLOv8 is more accurate than the other three methods in all defect detection. Two contradicting factors are speed and accuracy. Traditionally, the YOLO series is famous for its fast detection speed, while Faster R-CNN loses some speed to improve detection accuracy. However, in Table 7, YOLOv8 performs at a high accuracy rate while ensuring the speed of detection, which has a mAP of 0.96 and an inference time of 0.066; the detection time is 58.9%, 376.8% and 123.2% faster than YOLO v5, DETR, and Faster R-CNN. Compared with DETR, YOLOv5 and Faster R-CNN, YOLOV8 also has the highest mAP value, 0.96; however, YOLOv5 and Faster R-CNN only have 0.896 IPS and 0.125 IPS. DETR have better performance, but DETR have slower inference time, needing 0.267 IPS for per inference.

When testing, we used ground truth as a reference, and at the same time provided labels below the image for each bbox. From the detection results, these four object detection algorithms realized the automatic classification and localization of the rail internal defects

in B-scan images to further compare the performance of the detection results of these three networks. From typical detection results, YOLOv8 could precisely find the defects and outperform YOLOv5, DETR and Faster R-CNN in identification and classification, which benefits from feature fusion and a PAN structure. In contrast, DETR, YOLOv5, and Faster R-CNN have some problems. Among them, Faster R-CNN performs poorly, with a problem of detection errors and missed detection, especially in detecting fishbolt hole cracks and rail web cracks. When the defect appeared in the rail web area, Faster R-CNN could not identify the true defect or even identify the normal structures as defects. Compared to that, DETR does better in web defects detection, but missed detection still happens in bolt hole detection, YOLOv5 has better performance, but still has some problems in crescent-shaped fatigue crack detection. YOLOv8 can more accurately identify defects in the rail web and fishbolt hole crack area. However, there is no huge problem in YOLOv8. Instead, it finds the most appropriate prediction bbox. While in the detection of multiple defects with interference, when the defect appears in the rail connection area or nears a normal rail structure, Faster R-CNN could not identify the true defect from the interference terms and YOLOv8 performs better but sometimes with detection errors. In contrast, YOLOv8 could distinguish various categories of defects with a high detection rate and accuracy. Apart from that, the B-scan images of cracks around the bolt hole are covered or merged by the images of holes, leading to deformation, which is not easily distinguished. In the detection of bolt holes, YOLOv8 could not identify any defect; However, YOLOv8 detects the correct defects, which is similar to the detection results of the VOC dataset, which is that YOLOv8 has superior detection performance for obscured objects compared to DETR or YOLOv5.

### 5.2. Comparison of Defects Categories and Model Structures

To dig out the potential reasons for the different performances of these three models in our results, the processing differences of the models were analyzed according to the characteristics of the different defects. The feature fusion method in YOLOv5 and YOLOv8 is used for detecting small and dense objects, and mostly B-scan images of defects belong to small objects except for head defects, so that the YOLO series is superior to Faster R-CNN. Unlike the FPN layer of DETR, YOLOv5 adds a bottom-up feature pyramid behind the FPN layer, YOLOv8 also employs the FPN and utilizes anchor-free detection, which significantly accelerates the model's operational speed and achieves enhanced detection performance. With this combination, the FPN layer conveys powerful semantic features from the top-down and the feature pyramid conveys powerful localization features from the bottom-up, aggregating parameters from different backbone layers to different detection layers, which further improve the feature extraction. Furthermore, the mosaic data enhancement method as an improvement (trick) in YOLOv8 referenced the CutMix data enhancement method, which reads four images at a time, then flips, scales and cuts the four images, respectively, and arrange them in four directional positions to form a new image. The method enriches the detection background, solves the problem where the target scale in the datasets is too singular [30], and further enhances the localization capability of the model. However, the bag of specials (BoS) in YOLOv8, like SPP, Spatial Attention Module (SAM), Mish activation functions, and GIOU-loss, improve the detection accuracy of the model but add a little inferential cost; the training speed of YOLOv8 is slightly faster than that of YOLOv5 (4 h vs. 7 h). Figure 10 shows the contents of the four categories of defects in the B-scan images of the damage identified by the YOLOv8 algorithm: (a) rail head defects, (b) rail web, (c) head and bothhole defects, and (d) base defects, identified by the YOLOv8 algorithm in the B-scan images.

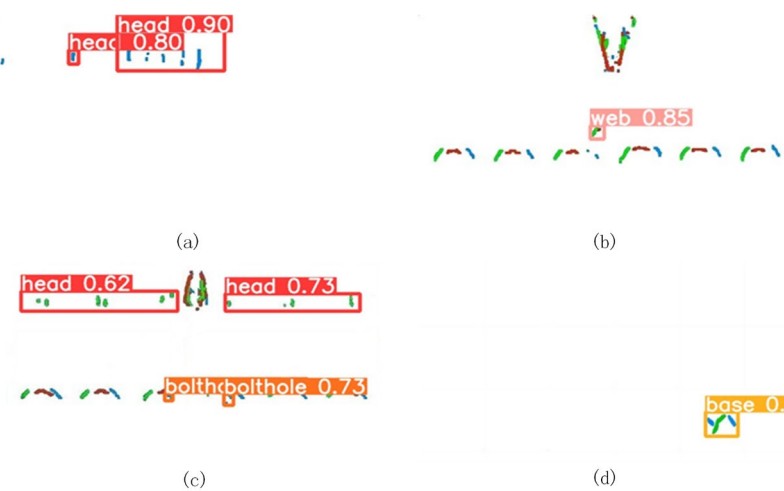

**Figure 10.** An example of YOLOv8 detection results.

## 6. Conclusions

We proposed deep learning-based approaches (object detection) to process rail internal B-scan images obtained by ultrasonic detectors to automatically identify and localize rail internal defects. Instead of conventional artificial secondary replay recognition, the new technique reduces labor and time costs and also meets both detection speed and accuracy requirements.

In the context of rail defect detection, the methodology proposed in this paper serves as an efficacious complementary approach. It capitalizes on the data provided by the GCT-8C rail flaw detector, applying a series of processing techniques to these data. As a result, the approach achieves commendable outcomes in detecting rail damage. This method enhances the existing diagnostic capabilities by leveraging the strengths of the GCT-8C instrument's data, which, when combined with the sophisticated data processing steps outlined in this study, significantly improves the accuracy and reliability of rail defect detection.

In this study, the representatives of one-stage deep learning networks, DETR, YOLOv5, and YOLOv8, and of two-stage deep learning networks, Faster R-CNN are proposed to detect four categories of internal defects, including crescent-shaped fatigue cracks (head), fishbolt hole cracks (bolt hole), rail web cracks (web) and rail base transverse cracks (base). After data pre-processing, these networks were transfer-learned based on B-scan images, the parameter was optimized, and the generalization capability was validated by adjusting hyperparameters and network details, then further tested and compared in different evaluation conditions. Compared to the Faster R-CNN model (mAP: 0.585, inference time: 0.125 IPS) with the problem of frequently missed and wrong detection, the YOLO series can detect almost all categories of defects in B-scan images. DETR demonstrates commendable performance across multiple metrics, including mAP, precision, recall, and the *F*1 score. Its overall performance closely approximates that of YOLOv8. However, in terms of single inference time, YOLOv8 (0.056 IPS) possesses a definitive advantage, evidencing superior efficiency in processing speed compared to DETR (0.267 IPS). There comes the first conclusion that the overall performance of YOLO series models is superior to that of Faster R-CNN on our datasets. Further, the YOLOv8 model further improves accuracy (mAP: 0.96 vs. 0.757 vs. 0.896) and inference time (0.056 IPS vs. 0.089 IPS vs. 0.067) over DETR and YOLOv5, and can accurately distinguish the interference items in B-scan images such as the image of bolt holes and surrounding cracks. There comes the second conclusion that the optimization algorithms (tricks) of YOLOv8 work well in our small dataset, finally achieving a balance between FPS and Precision. Last but not least, the two one-stage object detection networks are trained and tested at different input resolutions. YOLOv8 has a higher mAP value than our datasets. To conclude, the YOLOv8 model with

input sizes of 608 pixels × 608 pixels, with the highest accuracy and fastest detection speed, is the most suitable model for real-time rail internal B-scan image analysis.

For future work, unnecessary convolution layer structures in the deep learning network are suggested to be removed to reduce the calculation volume for B-scan image identification, and the specific operations in the network affecting the effectiveness of detection must be further explored. In addition, the object detection model cannot evaluate defect severity. Future research should focus on quantitative and qualitative studies of these defects, and segmentation algorithms might be used in the future.

**Author Contributions:** Conceptualization, S.W.; methodology, J.P. and W.H.; software, X.W.; validation, X.X.; formal analysis, W.H.; investigation, X.X.; resources, W.W.; data curation, S.W.; writing—original draft preparation, S.W. and W.H.; writing—review and editing, J.P.; visualization, Y.Z.; supervision, B.Y.; project administration, W.W. and B.Y.; funding acquisition, B.Y. and W.W. All authors have read and agreed to the published version of the manuscript.

**Funding:** This research was funded by the High-Speed Railway Infrastructure Joint Fund of the National Natural Science Foundation of China (No. U1734208); the National Natural Science Foundation of China (No. 52178442); the National Natural Science Foundation of China (No. 52278470); the Hong Kong Polytechnic University Postdoctoral Fellowship Scheme (No. 1-W29R).

**Institutional Review Board Statement:** Not applicable.

**Informed Consent Statement:** Not applicable.

**Data Availability Statement:** The raw data supporting the conclusions of this article will be made available by the authors on request.

**Conflicts of Interest:** The authors declare no conflicts of interest.

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
