# Peer review of "Automated Identification and Localization of Rail Internal Defects Based on Object Detection Networks"

_applsci, doi:10.3390/app14020805_

Round 1
Reviewer 1 Report
Comments and Suggestions for Authors
Reviewer finds the idea of implementing deep learning to solve a novel rail track defect problem. The paper is well-suited and commendable. However, the following should be considered to improve the paper.
1. Authors claimed to propose novel deep learning methods to process B-Scan images for automatic identification and localization of track defects. However, an expert reviewer does not seem to agree. YOLO v3, YOLO v4 and Faster RCNN as used in this paper are not novel deep learning and not proposed by the authors.
2. I think authors should rather say "This paper adopted state-of-the-art deep learning algorithms for novel B-scan images for automatic, identification and localization of rail internal tracks.
3. The most fundamental correction needed in this paper is section 3.1 "YOLO series detectors". Authors can not stop at YOLO v4 for a paper to be published in 2023 or 2024. Currently, there is YOLOv8. It is suggested that the paper should include YOLOv8 or a higher version if any. YOLOv4 as the highest in this paper makes the paper old and outdated.
Other corrections about abbreviations and little English corrections are in the attached PDF.
Inconsistent abbreviation and need for little English corrections as indicated in the attached PDF.
an example is "automatically" should be automatic (see abstract)
Reviewer 2 Report
Comments and Suggestions for Authors
This suggests a deep learning approach for the identification of rail defects in ultrasonic B-scan images and compares the performance of YOLOv4, YOLOv3, and Faster R-CNN.
1. The work is interesting, however, the authors need to discuss in detail the problem being solved by this method, and what its position is in relation to the GCT-8C rail defect detector. Is it a complementary or an alternative?
2. Since GCT-8C rail defect detector is a major reference, the authors must include a discussion on it in the results section
3. The data processing section is hard to understand, it is advised to include visual illustrations
4. To support the real-time claims it is important to present the computational time results
Round 2
Reviewer 1 Report
Comments and Suggestions for Authors
The authors have addressed all issues. Paper can be accepted.
Author Response
Dear Reviewer,
We would like to express our heartfelt thanks for the time and effort you dedicated to reviewing our paper. Your insightful feedback and constructive suggestions have been invaluable in enhancing the quality of our work.
Your meticulous attention to detail and expert guidance have significantly contributed to the refinement of our paper. We are grateful for your positive evaluation and recommendation for acceptance. Your support and encouragement are greatly appreciated.
Thank you once again for your valuable contribution to our work. We look forward to possibly collaborating or crossing paths in academic circles in the future.
Best Regards,
Wenbo

Reviewer 2 Report
Comments and Suggestions for Authors
The authors significantly improved the quality of the presentation, however, it is advised to address these comments
1. Improve the quality of figure 10.
2. clarify the time unit in the table (seconds)
3. Address other forms of damage detection such as Foucault-Current based methods " Optimal Axial-Probe Design for Foucault-Current Tomography: A Global Optimization Approach Based on Linear Sampling Method"and vibration-based methods "Experimental sensitivity analysis of sensor placement based on virtual springs and damage quantification in CFRP composite"
Comments on the Quality of English Language
No comment
